# Transition to a virtual model of physiotherapy and exercise physiology in response to COVID-19 for people in a rural Australia: Is it a viable solution to increase access to allied health for rural populations?

**Serene S. Paul[1], Angela Hubbard[2], Justin Johnson[2], Sarah M. Dennis[1,3]***

**1** Faculty of Medicine and Health, Sydney School of Health Sciences, Discipline of Physiotherapy, The University of Sydney, Sydney, New South Wales, Australia, **2** Phyz X, Orange, New South Wales, Australia, **3** Ingham Institute of Applied Medical Research, Liverpool, Australia

* sarah.dennis@sydney.edu.au

## Abstract

Virtual healthcare has the potential to increase access to allied health for people living in rural areas, but challenges in delivery of such models have been reported. The COVID-19 pandemic provided an opportunity for a rural practice of physiotherapists and exercise physiologists to transition service delivery to a virtual model of care which utilised a combination of phone, video, an exercise app and/or paper handouts. This study aimed to evaluate the uptake and outcomes from virtual delivery of allied health services, and to describe patient and clinician experiences of the virtual model of care. A parallel convergent mixed methods study was conducted. De-identified data from patients who were offered the virtual service between 15 March 2020 and 30 September 2020 were extracted from the database of the rural practice, as were data from patients attending the practice in-person during the same time in 2019 to serve as a historical comparison. De-identified data from a monthly survey tracking clinician experiences of delivering care virtually was also obtained from the practice. Quantitative data were presented descriptively. Between-group differences were compared using independent samples t-tests, and within-group longitudinal changes compared using paired t-tests. Semi-structured interviews were conducted among a purposive sample of patients using the virtual service, and focus groups conducted among clinicians providing this model of care. Qualitative data were recorded and transcribed verbatim, then thematic analysis conducted. During the study period, the practice delivered 4% (n = 242) consultations virtually. Thirty-seven of the 60 patients (62%) using the virtual service were new referrals. Patients attended fewer sessional appointments virtually and a smaller proportion of patients reported high satisfaction with virtual care, compared to those who received in-person care the previous year (p < .05). Clinician confidence in delivering virtual care did not change significantly over time (p>.05), though clinicians not providing virtual care in a given month perceived their lower confidence than those who did provide virtual care (p < .05). Five themes influencing the success of virtual allied health provision emerged from patient

**Data Availability Statement:** All relevant data are within the paper and its Supporting information files.

**Funding:** This study was supported by a Charles Perkins Centre (University of Sydney) Early Mid-Career Researcher 2020 Seed Funding grant (SP). The funders had no role in study design, data collection and analysis, decision to publish, or preparation of the manuscript.

**Competing interests:** Serene Paul and Sarah Dennis have declared that no competing interests exist. Phyz X and Phyz X 2U constitute the practice reported in this study. We have read the journal's policy and the authors of this manuscript have the following competing interests. Angela Hubbard is a co-owner of Phyz X 2U. Justin Johnson owns Phyz X and co-owns Phyz X 2U. This does not alter our adherence to the PLOS ONE policies on sharing data and materials.

interviews and clinician focus groups: adaptation of program elements for virtual delivery, conduct of virtual treatment, clinician flexibility, patient complexity and communication. The theme of communication influenced all the other themes. Virtual healthcare is a potential solution to address lack of access to allied health practitioners in rural areas, but may not suit all patients. Establishing a therapeutic relationship and ensuring people have access to adequate resources prior to virtual care delivery will optimise successful adoption of virtual care models. A hybrid model incorporating limited in-person consultations with virtual consultations appears a more viable option.

## Introduction

Allied health interventions, such as physiotherapy and exercise physiology, are effective for improving health outcomes for a range of musculoskeletal conditions [1–4] and chronic diseases [5], however, access to allied health is not universally available to all patient groups particularly disadvantaged populations living in rural/remote areas [6, 7]. Virtual healthcare, ie, healthcare delivered via telephone or video, has the potential to improve access to allied health services for people living in rural areas [8]. However, some allied health providers such as physiotherapists and exercise physiologists traditionally provide hands-on therapy which could make the transition to a virtual model of care challenging. Virtual healthcare models have been shown to have comparable outcomes to traditional in-person service delivery models for a range of musculoskeletal [9, 10] and chronic conditions [11, 12], but challenges in virtual delivery persist [13, 14], with conflicting reports about adherence [9, 10, 13]. Factors including poor digital literacy, poor mobile coverage in rural/remote areas and lack of access to technology [13–15] have been reported to negatively impact adherence to virtual healthcare. In Australia, a further barrier to virtual care was that most health insurance plans and the government funded Medicare system did not cover virtual delivery of physiotherapy or exercise physiology interventions, adding to the access issues for people in rural and remote areas.

In March 2020, in response to the COVID-19 pandemic, the Australian government and some private health insurance companies agreed to fund access to virtual models of physiotherapy and exercise physiology for some conditions [16]. On 23 March 2020, the New South Wales (NSW) state government introduced a 90-day period of lockdown to control the spread of the virus. During this time, allied health services, especially outpatient services and those in community and primary health, either transitioned to a virtual model of care or offered a limited in-person clinical service.

Prior to the COVID-19 pandemic, an allied health provider in rural NSW provided physiotherapy and exercise physiology services through a clinic, or a mobile outreach service supplemented with virtual health coaching for chronic disease management. The COVID-19 pandemic required providers in both rural and metropolitan settings to offer patients their physiotherapy and exercise physiology services through a new service delivery model using only virtual health. For some providers, including the rural allied health provider studied herein, patients who lived close to the clinic and preferred to use the traditional in-person service during the pandemic could opt to do so. This enforced practice change provides an important natural experiment to evaluate the uptake, acceptability and outcomes of this change in service delivery on patient and service level outcomes in a rural area. The results will inform whether a fully virtual healthcare model may be a feasible solution to help address poor access to allied healthcare beyond the COVID-19 pandemic.

The aim of this study was to describe (i) the uptake and outcomes from the virtual delivery of a rural model of allied health in response to COVID-19, and (ii) how virtual service delivery impacted patients and clinicians, including their experience of the virtual model of care.

## Material and methods

We conducted a parallel convergent mixed methods study [17] comprising an observational study with a historical comparison and qualitative interviews using a phenomenological approach with patients and clinicians. The setting was a private physiotherapy and exercise physiology practice in two rural towns (population of approximately 42,000 to 45,000) in NSW, Australia. The practice usually provides in-person physiotherapy and exercise physiology services either in the clinic through sessional appointments, or a 12-week program via a mobile outreach service to satellite communities over an approximate area of 785,375 km$^2$ supplemented with virtual health coaching for chronic disease management. In-person services began with an initial consultation which comprised a thorough assessment followed by intervention as determined by the clinician, this could include provision of advice/education, exercise and/or manual therapy; subsequent follow-up consultations were booked as required. Follow-up consultations comprised a brief assessment followed by intervention as indicated. The patient was discharged from the service when the patient and treating clinician determined that no further follow-up was required (i.e., sessional service). The mobile outreach service had a similar in-person initial consultation as the in-person services, with the final evaluation 12 weeks later following the same format. During the intervening 10 weeks, intervention was in the form of weekly health coaching delivered virtually to update and progress exercise programs and provide health advice/education. All consultations were 30 minutes in duration except for the mobile outreach service's initial consultation which was 60 minutes. During the partial lockdown period of the COVID-19 pandemic the practice introduced a fully virtual health model and ceased the mobile outreach service, though those who lived close to the two rural towns were able to access the traditional in-person service delivery model, as healthcare was exempt from COVID-19 lockdown restrictions in Australia. The fully virtual model followed the format of either the sessional in-person service or the 12-week mobile outreach service, depending on the patient's preference. The platforms used depended on the patient's preference and access to technology, and varied from phone to freely available video-conferencing software, with or without use of an exercise app (Physitrack, London, UK). The practice did not have formal technology support, hence any troubleshooting was conducted by the clinician prior to or during the consultation.

The study received ethical approval from the University of Sydney Human Research Ethics Committee (reference: 2020/412). De-identified clinical data was obtained under a waiver of consent. Patients provided verbal informed consent to participate in the interviews, while the clinicians provided written informed consent to participate in the focus groups. The outcomes for this mixed methods study were to describe: 1) Uptake of the virtual service, compared to the number who accessed traditional in-person services, during the study period; 2) The characteristics of patients who accessed the virtual service; 3) The proportion of patients who reported being satisfied with the virtual service compared to historical controls who used traditional in-person services during a similar time period; 4) Clinician confidence to deliver telehealth; and 5) Patient and clinician experiences of virtual healthcare uptake and delivery, respectively.

The observational study used de-identified clinical data extracted from the practice database for all patients who were offered and received the virtual service between 15 March 2020 and 30 September 2020. Historical de-identified data for patients attending the traditional

service delivery model during the same period in 2019 were also extracted to serve as a comparison. The same number of patients who attended each service subtype (sessional or mobile outreach) virtually was matched with patients in the historical cohort. Additionally, patients were matched on decennial age group, sex and presenting problem (spinal pain, lower limb musculoskeletal problem, upper limb musculoskeletal problem, neurological problem and other problems), though owing to the small numbers in the final sample we were only able to match patients with a minimum of two historical controls per presenting problem category. The service level data collection has been piloted in a previous research project conducted by the research team [13]. The de-identified data extracted for both time periods included: patient number, date of consultation, client type (existing or new), service type (virtual or in-person), referral to service, consult type (baseline assessment, follow-up consult), age, sex, place of residence (classified as metropolitan, inner regional, outer regional, remote, very remote; the latter four are rural areas with progressively smaller populations and greater distances from major population centres) [18, 19], presenting problem, past medical history, goal for therapy, and rating of overall program satisfaction (0–5 rating, where 0 = poor and 5 = excellent).

At the start of the lockdown, clinicians (physiotherapists and exercise physiologists) completed a monthly survey to track their confidence in delivering virtual health services. De-identified data extracted from this survey included: clinician number; prior training in providing virtual care (yes/no); month the survey was completed; number of virtual consultations conducted in the past month; methods used to provide virtual care; confidence in delivering a physiotherapy/exercise physiology initial assessment or intervention online (0–10 rating scale, where higher scores indicate greater confidence); and self-rating of ability to diagnose the patient's problem, ability to provide an intervention, communication skills and ability to problem solve technical challenges virtually compared to in-person (0–10 rating scale, where higher scores indicate greater ability).

Semi-structured interviews with patients who had taken part in the virtual program were conducted between August 2020 and January 2021; each interview lasted approximately 45 mins. A purposive sample of patients were invited to participate by the practice administrative team. After each interview, the researchers (SP and SD) discussed the key findings and reflected on the type of patients to be approached for interview to ensure breadth and depth in the data. For example, early patients were long-time patients of the practice who talked about continuity of care, so we wanted to ensure we interviewed new patients to understand the concept of continuity of care for people new to the service. Interviews ceased when data saturation was reached or no further patients consented to be interviewed. The interviews were conducted by the researchers (SP, SD) by telephone and digitally recorded. They were transcribed verbatim.

Focus groups were conducted with physiotherapists and exercise physiologists who provided virtual care during the study period; each focus group lasted approximately 60 mins. At the time of the study, there were seven clinicians working at the service who were invited to participate in two focus groups; by happenstance one group comprised physiotherapists and the other group the exercise physiologists. The focus groups were conducted using videoconferencing and were digitally recorded and transcribed verbatim. Member checking with the clinician participants was conducted by sharing the transcripts with them and seeking feedback on accuracy.

Supplementary material 1 in S1 File contains the interview and focus group guides. Rigour of the qualitative component was informed by the methods of Forero et al [20]. Credibility was assured by ensuring that the interviews were conducted by researchers experienced in qualitative research (SD and SP) and there were debriefs after each interview or focus group between these two researchers. During the debrief, the researchers also reflected on the characteristics

of the participants which informed the purposive recruitment which continued until data saturation was reached.

## Data analysis

The de-identified quantitative data were analysed descriptively using SAS v9.4 (SAS Institute, Cary NC) statistical software. Results were reported as number (%) for describing proportions and mean (SD) for describing continuous data, as these were normally distributed. Missing data were reported as such, except where program satisfaction at follow-up was missing in which case this was reported as 'no change'. Changes between the virtual service and historical comparisons were analysed using independent samples t-tests. Post-hoc differences in the mean number of consultations delivered by the virtual service compared to traditional services were analysed using an analysis of variance to examine interactions between time (i.e., year) and service subtype (i.e., sessional vs mobile outreach). Changes in clinician confidence over the study period were analysed using paired t-tests. The clinician survey data were also aggregated according to whether individual clinicians had provided virtual consultations in any given month or not and compared using independent samples t-tests.

The transcripts of the interviews and focus groups were read and coded inductively line by line with the use of NVivo v12 (QSR International Pty Ltd, 2018). Thematic analysis was guided by the methods of Braun and Clark [21]. The coding and emerging themes were discussed within the research team (SP, SD, AH), this process was iterative and the coding and emerging themes updated following discussions.

Finally, the results of the quantitative and qualitative data were merged to inform and enrich the findings. The qualitative results provided explanation for some of the quantitative findings and enabled the researchers to develop a more detailed understanding of the experiences of virtual care. Data from the patient and clinician interviews were triangulated. The research team included experienced quantitative and qualitative researchers (SD, SP) and clinicians with experience of delivering virtual and in-person services and this may have influenced the interpretation of the data.

## Results

### Quantitative findings

Between 15 March 2020 and 30 September 2020, there were a total of 90 patients who expressed interest in accessing the virtual service, and of those 50 (56%) were new referrals. Of these 90 patients, 30 (33%) chose to wait for the resumption of in-person services and 60 (67%) took up the virtual service. The latter included 37 (62%) new referrals to the virtual service, while the remainder were already attending the standard service and chose to transition to the virtual service during the COVID-19 lockdown. These 60 patients attended a total of 242 consultations over the study period, a mean of 5.0 (SD 3.2) per patient. In comparison, the practice delivered a total of 5958 traditional in-person consultations over the same period to 1267 patients, including physically distanced group classes. Virtual services peaked in March and April 2020 then quickly dropped off in June as the NSW government began lifting COVID-19 restrictions. The increase in traditional in-person services mirrored the drop-off in the virtual service (Fig 1).

Table 1 describes characteristics of the 60 patients who took up the virtual service. Twenty-nine of the 60 patients (12% of the 242 consultations) had initial assessments through the virtual service. Most patients saw a physiotherapist only (33/59, 56%), with similar numbers seeing either an exercise physiologist only (14/59, 24%) or both (12/59, 10%); data was missing

## Number of consultations by service type

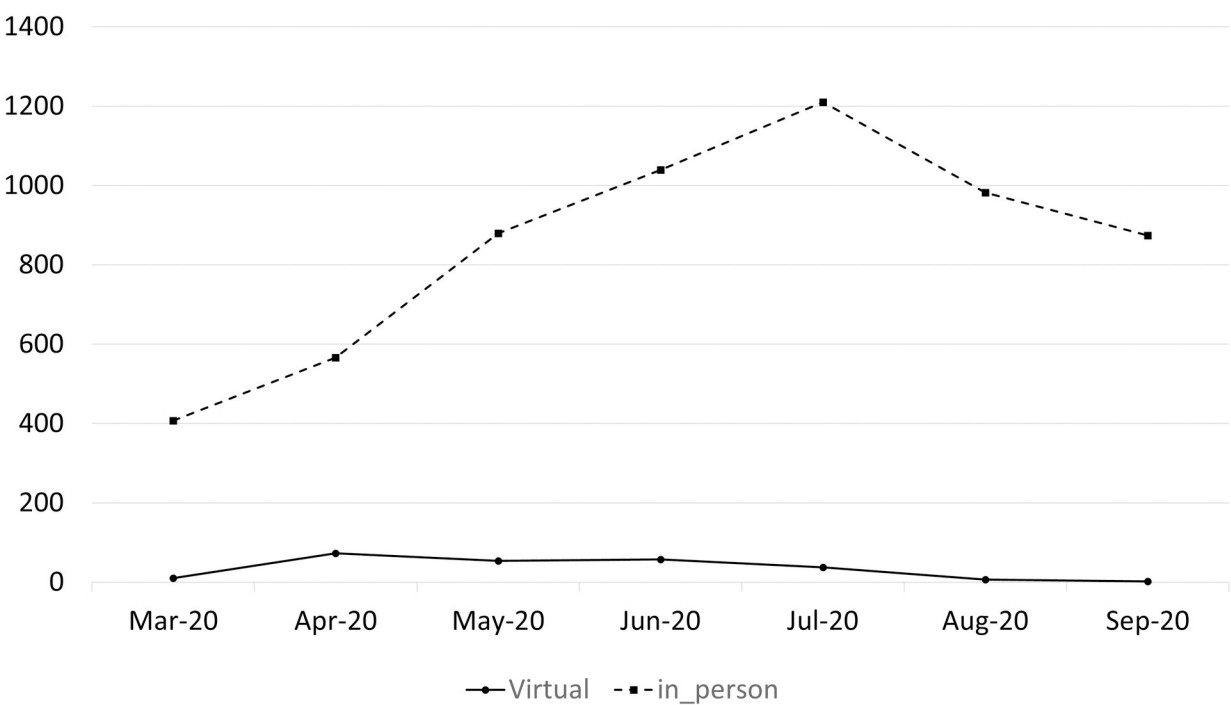

**Fig 1. Number of consultations by service type, between 25 March 2020 and 30 September 2020.**

for one patient. Each patient attended a mean of 4.0 (SD 2.9, range 1–11) virtual consultations over the study period.

Clinical records with details of patients' presenting problems, comorbidities and treatment goals were only available for 32 (53%) of the 60 patients (Table 2). Of these 32 patients, 30 (94%) set goals for treatment over the study period. Program satisfaction data were available for 22/32 (69%) patients. Overall, participants reported a slight reduction in ratings of program satisfaction during the virtual service (mean change -0.4 on a 5-point rating scale, SD 1.1, range -2 to 2). Of these 22 patients, 3 (14%) reported an increase in program satisfaction with the virtual service, 11 (50%) reported no change in program satisfaction and 8 (36%) reported reduced satisfaction.

Compared to historical controls who accessed in-person services over a similar period in 2019 (Table 2), there were no significant changes to the overall number of appointments attended (p = .50) or in the mean program satisfaction rating (p = .55). However, closer examination of the data revealed two interesting findings. The number of virtual consultations for sessional appointments dropped significantly compared to the traditional model (overall model p = .03); this change was driven by the interaction of year and service subtype (p = .004). There was a smaller proportion of patients who reported higher program satisfaction and a greater proportion who reported a reduction in program satisfaction with the virtual service, compared to the traditional model (p = .02).

All seven allied health clinicians delivered between 6 to 42 (mean 20.9, SD 13.3) virtual healthcare consultations over the study period. Clinicians delivered virtual healthcare primarily through video and/or phone (93% and 85%, respectively, across the study period); the

**Table 1. Characteristics of patients who accessed the virtual service between 25 March 2020 and 30 September 2020.** Data reported as n (%); or mean (SD), range.

| Characteristic | | Attended the virtual service | Registered with the virtual service but did not attend |
|---|---|---|---|
| Number of patients | | 60 | 30 |
| Sex[†] | M | 13 (22%) | 3 (21%) |
| | F | 45 (78%) | 11 (79%) |
| Age (years)[‡] | | 46.2 (16.7), 9–86 | 35.3 (16.4), 15–67 |
| Place of residence[§] | | | |
| Metropolitan | | 2 (3%) | 0 (0%) |
| Inner regional | | 50 (83%) | 22 (88%) |
| Outer regional | | 3 (5%) | 2 (8%) |
| Remote | | 4 (7%) | 1 (4%) |
| Very remote | | 0 (0%) | 0 (0%) |
| New referrals | | 37 (62%) | 13 (43%) |
| Health professional | | 9 (24%) | 1 (8%) |
| Workplace | | 10 (27%) | 2 (15%) |
| Self-referral | | 11 (30%) | 6 (46%) |
| Other | | 3 (8%) | 4 (31%) |

[†]missing sex in n = 2/60 (3%) who attended the virtual service or n = 12/30 (40%) registering for the virtual service.
[‡]missing age in n = 3/60 (5%) who attended the virtual service or n = 15/30 (50%) registering for the virtual service.
[§]missing place of residence in n = 1/60 (2%) who attended the virtual service or n = 5/30 (17%) registering for the virtual service. Inner regional, outer regional, remote and very remote refer to rural areas with progressively smaller populations and greater distances from major population centres (18, 19).
^unknown referral source in n = 5/37 (14%) new referrals who attended the virtual service.

mode was determined by the technology available to the patient. While all clinicians delivered the highest volume of virtual services across April and May, not all clinicians continued to deliver virtual services from June onwards, consistent with the reducing volume of virtual consultations. All clinicians received training in how to deliver healthcare virtually, with most receiving this training in March and April 2020, though two therapists had 2–3 years' prior experience in delivering healthcare virtually. Overall, clinicians felt confident in assessing and treating patients virtually (Table 3). They similarly felt confident in their communication skills and problem-solving abilities to troubleshoot technical problems to effectively deliver care virtually. However, they were far less confident in diagnosing a patient's problem or providing an intervention virtually compared to an in-person consultation. While clinicians perceived their ability in diagnosing and providing intervention virtually improved slightly over the study period, confidence reduced slightly overtime, though none of these changes were statistically significant ($p > .05$). When probing reasons why this may be, we discovered that clinicians who did not provide a virtual consultation in a given month perceived their ability and confidence lower than those who did provide virtual consultations ($p < .05$, Table 3).

## Qualitative findings

Five themes which influenced the success of virtual allied health consultations were identified from the interviews of patients and clinicians. These themes were: program adaptation for virtual delivery, conduct of virtual treatment, clinician flexibility, patient complexity and communication. The theme of communication influenced all the other themes.

**Table 2. Clinical characteristics of patients accessing the virtual service (March–September 2020) and historical controls (March–September 2019).** Data reported as mean (SD), range; or n (%). Between-group differences were analysed using independent samples t-tests.

| Characteristic | Virtual service (2020, n = 32) | In-person service (2019, n = 32) | Between group difference (p-value) |
|---|---|---|---|
| Number (%) new referrals | 19 (59%) | 15 (47%) | |
| Number of appointments attended | 5.0 (3.2), 0–11 | 5.8 (5.0), 2–31 | .50[a] |
| Sessional | 3.2 (2.0), 1–7 | 7.1 (6.8), 2–31 | .03* |
| Mobile outreach[b] | 6.6 (3.2), 0–11 | 4.5 (2.2), 2–8 | |
| Reasons for seeking therapy | | | |
| Spinal pain | 12 (38%) | 11 (34%) | |
| Lower limb musculoskeletal problem | 5 (16%) | 10 (32%) | |
| Upper limb musculoskeletal problem | 2 (6%) | 5 (16%) | |
| Other[c] | 13 (41%) | 6 (19%) | |
| Treatment goals | | | |
| Improve strength | 16 | 9 | |
| Improve physical activity, exercise and/or wellbeing | 16 | 14 | |
| Improve aerobic fitness | 4 | 1 | |
| Weight loss | 5 | 5 | |
| Reduce pain | 3 | 14 | |
| Improve flexibility | 6 | 10 | |
| Improve balance | 0 | 1 | |
| Return to work | 2 | 1 | |
| Injury prevention | 2 | 0 | |
| Improve mental health | 1 | 1 | |
| Smoking cessation | 0 | 1 | |
| Past medical history | | | |
| Spinal pain | 9 (28%) | 14 (44%) | |
| Lower limb musculoskeletal problem | 8 (25%) | 10 (31%) | |
| Upper limb musculoskeletal problem | 4 (13%) | 3 (9%) | |
| Neurological problem | 3 (9%) | 2 (6%) | |
| Other[d] | 8 (25%) | 3 (9%) | |
| Number reporting program satisfaction | 29 (91%) | 30 (94%) | |
| Rating of program satisfaction (0–5)[e] | 2.7 (1.4), 1–5 | 2.9 (1.1), 1–5 | .55[f] |
| Change in program satisfaction | -0.3 (0.9), -2 to 2 | 0.4 (1.1), -1 to 3 | .02[g] |
| Satisfied with program | 3 (14%) | 12 (40%) | |
| Satisfaction unchanged[h] | 11 (50%) | 12 (40%) | |
| Dissatisfied with program | 8 (36%) | 6 (20%) | |

*The model examining virtual vs traditional services by year, service subtype (ie, sessional vs mobile outreach), and the interaction of year and service subtype. Overall model p = .03. Individual effects: year: p = .37, mean change 0.9 (95% CI -1.0, 2.9); service subtype: p = .67, mean change -0.4 (95% CI -2.4, 1.6); interaction of year and service subtype: p = .004.

[a]Change from 2020 (virtual service) compared to 2019 (in-person service): mean 0.7 (95% CI -1.4, 2.8) reduction.

[b]The 12-week mobile outreach service has a fairly consistent number of consultations (median 5–7) [13], while those from the in-clinic sessional service varies widely. Note that the low mean number of mobile outreach appointments in 2019 likely reflects a proportion of participants who had recently commenced or were partway through their program when the historical analysis period ended.

[c]Includes vertigo, pain, headache or no reason.

[d]Includes mental health, intellectual disability, gastro-intestinal issues, sleep apnoea or no reason.

[e]Higher scores indicate higher level of wellness and program satisfaction.

[f]Change from 2020 (virtual service) compared to 2019 (in-person service): mean 0.2 (95% CI -0.5, 0.9) reduction.

[g]Change from 2020 (virtual service) compared to 2019 (in-person service): mean 0.7 (95% CI 0.1, 1.3) reduction.

[h]n = 5 with satisfaction unchanged were due to missing data at follow-up.

**Table 3. Clinician (n = 7) confidence for delivering healthcare virtually over the study period.**

| Characteristic | Change over time | | | Differences between clinicians who did or did not deliver virtual services in a given month | | |
|---|---|---|---|---|---|---|
| | Study start (April 2020) | Study end (September 2020) | change over time (mean difference (95% CI); p-value) | Saw patients virtually within any given month (n = 27 responses) | Did not see patients virtually within any given month (n = 10 responses) | between-group difference (mean (95% CI); p-value) |
| Confidence in delivering an initial assessment online | 8.3 (1.0), 7–10 | 8.0 (1.6), 5–10 | -0.29 (-1.56, 0.99); p = .60 | 8.7 (1.1), 7–10 | 7.5 (1.2), 5–9 | -1.24 (-2.06, -0.42); p = .004 |
| Confidence in delivering an intervention online | 8.4 (1.3), 7–10 | 7.9 (1.6), 5–10 | -0.57 (-1.75, 0.61); p = .28 | 8.6 (1.4), 4–10 | 7.3 (0.9), 5–8 | -1.29 (-2.28, -0.30); p = .01 |
| Ability to diagnose the patient's problem online | 4.3 (1.8), 3–8 | 5.0 (2.7), 2–10 | 0.71 (-1.11, 2.54); p = .38 | 5.0 (2.1), 2–10 | 5.3 (1.7), 3–7 | 0.26 (-1.26, 1.78); p = .73 |
| Ability to provide an intervention online | 4.9 (1.5), 4–8 | 5.9 (2.8), 2–10 | 1.00 (-1.39, 3.39); p = .35 | 5.2 (2.4), 1–10 | 5.0 (1.4), 3–7 | -0.19 (-1.86, 1.49); p = .82 |
| Effectiveness of communication skills for delivering a virtual service | 8.0 (1.2), 7–10 | 8.0 (1.9), 5–10 | 0 (-1.41, 1.41); p = 1.0 | 8.0 (1.3), 5–10 | 6.9 (1.5), 5–10 | -1.10 (-2.14, -0.06); p = .04 |
| Ability to problem solve technical challenges when delivering a virtual service | 7.6 (1.6), 5–10 | 7.6 (1.9), 5–9 | 0 (-1.31, 1.31); p = 1.0 | 7.8 (1.3), 5–10 | 6.4 (2.0), 4–9 | -1.38 (-2.51, -0.23); p = .02 |

Data reported as mean (SD), range. Ratings range from 0 to 10, with higher scores indicating greater confidence.

^The practice first surveyed clinicians in April 2020, approximately one month after first transitioning to virtual service delivery.

**Adaptation of program elements for virtual delivery.** This theme mainly focuses on the adjustments to the prescribed exercise program for virtual delivery. Patients missed the access to equipment at the clinic which limited the virtual exercise program and this may have impacted the lower satisfaction with the virtual program in the quantitative data. For example, the exercise program for one patient was adjusted to focus on the upper limb because of access to equipment: "*We mainly did upper body because–yeah, I didn't have the equipment at home*" (0F1).

Another patient needed to focus on lower body exercise after a brain injury but was not able to and the clinician felt this patient had declined during the virtual period: "*Normally we do quite a lot of heavy strength work and then we do his balance [. . .] his memory is terrible, but he remembered to tell me that he was really, really sore after the first session back, which I thought [. . . as the weight] wasn't that heavy, so he definitely likely declined a little bit in that time as well*" (Clinician FG2).

Patients reported that the clinicians also had to adapt balance exercises to ensure patients were safe when they exercised: "*She adapted it so when I was doing say some tandem walking in my kitchen, she made sure that I had–I always had something close to me that I could grab onto if I needed* to" (01F).

In spite of reporting high levels of confidence for conducting virtual assessments and interventions, there were challenges experienced by clinicians. They talked about not being able to use their hands to correct patients' exercise technique and form. Overall, this indicated that the exercise program implemented during the virtual period was not delivered at the same intensity as it would have been in-person which may explain the reduced satisfaction with virtual therapy reported by the patients in Table 2.

**Conduct of virtual treatment.** This theme links to and extends the program appropriateness theme. As with all virtual healthcare, both patients and clinicians indicated there were technical issues accessing care. This was partially alleviated by providing access to the Physitrack app in advance and opportunities to troubleshoot before the actual appointment.

Fortunately, those clinicians providing regular virtual care reported feeling confident with troubleshooting. Both patients and clinicians described that the virtual program took longer to work through than the in-person program: "*By the time it was explained, the exercises explained, it was much slower and you couldn't do as many exercises in the time*" (06M).

Then there were the delays in trying to position the device so that the clinician could monitor the exercise: "*I had to try and balance the computer somewhere that it could see what you were doing. It certainly didn't have the amount of exercise that you could fit in when it was face to face*" (06M).

Another patient described moving the therapist around her house: "*So, I would go into the kitchen and I would sit her up on–[laughs] I'd sit her up on my tea and coffee thing and then I would walk across the kitchen and I'd do like push-ups from the bench and so she could see me. She couldn't see all of me obviously*" (01F).

Again, this meant that patients were not receiving the same intensity of practice during the virtual program. For some patients, such as people with a disability or children, the delivery of the program was dependent on a carer who may or may not be able to help with the therapy: "*He has a bit more of an intellectual disability and behaviour issues as well [. . .] he's not motivated to exercise when I go there to his school, in his own environment and we have half an hour, let alone on a screen. He lives with his mum, but his mum's got heaps of other kids that she's running round looking after, so–I think it's just keeping them moving the whole time, for the whole half an hour. Whereas in the clinic I can just say tip, you're in and then they just start running*" (Clinician FG2).

Both patients and clinicians felt that it was more challenging to provide holistic management virtually: "*I mean I'd had a bad shoulder and I've had a broken wrist and various things as well, so she possibly couldn't identify those on Zoom whereas with face-to-face, the physio is there watching the whole time*" (05F).

**Patient complexity.** This theme relates to the patient factors that influence the success of virtual care. Overall patients were grateful to be receiving therapy virtually and able to continue with their exercise program but were keen to revert back to in-person services when able. There were some patient groups where virtual therapy seemed to work better than others; clinicians felt that it was not so good for older people, people with disabilities either intellectual or physical, or those with balance problems as a reason for presenting. Most of the patients interviewed appeared to have good health literacy and reported being engaged in their therapy and felt that without virtual therapy they would not do their exercises and they would deteriorate further: "*Yes, absolutely. I think the continuity was important. If we go–now I find, if we go a week without our exercise class, we tend to go back a bit*" (06M).

Those patients with an existing in-person relationship with the therapists tended to feel more confident exercising virtually as they had learnt the correct technique but some still worried whether they were doing the exercise correctly: "*So yes, the videos that showed you how to do them were excellent, but you know how sometimes when you're doing something you can't see if you're in perfect alignment unless you've got a mirror next to you*?" (04F).

New patients enrolled during the virtual period tended to have a better experience when they came with clear expectations and goals for their therapy: "*I'm very aware that I need to maintain the balance between my core and my back to allow myself not to succumb to a back injury. So I just wanted to maintain that and add on to any additional things that I was doing their suggestions, and also looking at flexibility maintenance and benefits of stretching. [. . .] So they were the two areas that were what I was looking for advice on to look for maintaining those and keeping good body strength*" (04F).

**Clinician flexibility.** This theme seemed to be important for the success of telehealth and those clinicians that delivered more virtual care were significantly more confident in their

abilities to assess, treat and adapt programs than those that did less virtual consultations. Clinicians needed to adapt to the virtual mode of delivery especially for conducting an initial virtual assessment, they talked about using different skills in their "*clinical toolbox*": "*So your assessment doesn't completely change, but certainly you wouldn't conduct the same assessment that you would on a telehealth as you do face to face*" (Clinician FG1). And "*Then you're going back to your clinical toolbox of what else can you use to try and diagnose an ACL? You've got to go off is there swelling, is there instability in there, can they weight bear on it, can they single leg stand, can they hop, can they jump, can they do a bridge, how well are their hamstrings activated and that sort of thing*" (Clinician FG1).

Clinicians described being more flexible in their approach to exercise prescription and adapting to the lack of equipment and safety already mentioned. They also talked about making changes to adapt to some of the technical challenges they were presented with: "*I had an older model laptop to start with as well and it did not work on Physitrack for video calls, so that was annoying. But then it was good, because I would just use my phone for the video and type my notes on my laptop, so it worked good in that way*" (Clinician FG1).

Generally, clinicians were enthusiastic about the transition to virtual care although most had had some previous experience of telehealth: "*Yeah, I mean I think that prior experience was really good and yeah, first consult [. . .] and then you finish and you're like oh, that was fun. It was easy. I don't think it changed at all, because at the end of the day when people are coming in you're still using your words to explain exercises and your body language as well, which they can see via Telehealth*" (Clinician FG1).

Overall, all the clinicians were positive about the experience although one did admit that they felt "*less invested in the client*" during virtual therapy.

**Communication.** This was a strong theme that overlapped with the other four themes. Clinicians found it easier to build rapport with a patient during an in-person consultation. This was reflected in those patients who had started in-person who felt they had developed rapport which helped during their transition to the virtual model: "*The level of connectedness you get when you're communicating with someone. I don't think you get it as well over telehealth. When you come in and you really are quite vulnerable when you present as a client and you're telling them all of these things and you do get quite deep in your subjective assessment with a new client. I think it's a lot easier to get to that point when you're talking to someone in a face-to-face environment*" (Clinician, FG1).

In contrast, for new patients, rapport was more challenging, particularly when there was difficulty scheduling virtual appointments: "*I never got past the sort of buy in stage really or got to actually talk to someone about what needed to be done to actually start the program*" (03M).

There were also opportunities to build rapport through generally "*catching up*" during the in-person model treatment time. Clinicians did not do this during virtual appointments, keeping the communication focused on the program being delivered as they found it more challenging to provide the verbal cues for exercise virtually and chatter would interfere with this. Clear communication skills were essential for verbal cues to describe how to perform the exercises correctly; during in-person therapy the clinician would be able to provide guidance using their hands: "*You have to be really particular with your verbal cue. In clinic when you ask someone to do something, if it's not quite right as manual therapists often you'll find you just put a little hand under and it just corrects. When you can't just adjust somebody, you have to be able to cue someone without making huge adjustments in what they're doing*" (Clinician FG1). And "*I talk the whole session with my clients, like 45 minutes of just catching up [. . .] You can't do that on telehealth*" (Clinician FG2).

The clinicians interviewed felt that those with strong communication skills would transition well to a virtual care environment unlike their colleagues who relied more heavily on a hands-

on approach and was reflected in the results of the clinician survey. There was some concern that newly graduated therapists might not have fully developed their communication and broader clinical skills sufficiently to be as effective delivering virtual care.

## Discussion

Although COVID-19 provided an opportunity to evaluate to what extent virtual therapy was utilised to access allied health care in rural and remote areas, we found that there was a much lower uptake of virtual healthcare in comparison to patients who chose to either continue with in-person care or wait until lockdown restrictions eased fully, as healthcare was considered exempt from COVID-19 restrictions in Australia. Only 4% of total consultations were delivered virtually by the practice over the study period. This lower uptake was also reflected in lower satisfaction scores with virtual care. The results of this small study suggest that while virtual healthcare is a potential solution to address lack of access to allied health practitioners such as physiotherapists and exercise physiologists in rural settings, it may not suit all patients [22].

Evidence from meta-analyses demonstrate comparable or better health outcomes from exercise interventions delivered virtually compared to usual care, particularly in cardiac [11] and musculoskeletal populations [11, 23], though when therapy dosage is considered, outcomes between virtual and in-person delivery are comparable [23, 24]. In contrast, populations with more complex needs (eg, neurological patients [11]) or those less familiar with technology (eg, geriatric patients [25]), do not appear to benefit from virtually delivered intervention. This was reflected in our qualitative findings where clinicians and patients alike felt that those with more complex conditions received more intensive therapy in-person. Clinically, it appears that triaging patients to offer virtual care to those with less complex needs and treating those with more complex needs in person may improve patient outcomes. We found that patients had lower levels of program satisfaction with virtual therapy compared to historical controls who accessed in-person services the same time pre-COVID-19, including some who accessed a hybrid in-person and virtually delivered model, in contrast with prior findings [10, 26]. Nevertheless in one study, despite reporting higher satisfaction with virtual health, patients who attended in-person services received better outcomes [10]. While we were unable to report on health outcomes following therapy in our study, the low uptake of the virtual model compared to the high volume of patients still accessing in-person services despite COVID-19 restrictions highlighted that people who valued in-person services still elected this model of care over a virtual model [25, 27]. This is consistent with prior findings of chronic pain patients' preferences for healthcare delivery, which found that people do not like healthcare delivered completely virtually [27], instead preferring in-person services or a hybrid model incorporating virtual healthcare with some in-person sessions with their healthcare provider [27], the latter associated with good patient adherence and health outcomes [13].

Our qualitative results highlighted the importance of establishing a therapeutic relationship for virtually delivered healthcare to succeed. This is an important and new finding, as prior studies demonstrating the benefits of virtually-delivered exercise programs assessed physical outcomes in-person at baseline prior to delivery of the virtual intervention. These prior studies thereby provided an opportunity to establish a therapeutic relationship [10, 11, 23, 28], unlike our study where both assessments and intervention were delivered virtually. This result also likely explains our finding of a substantial reduction in the number of sessional consultations delivered virtually; note that the fewer in-person consultations during 2019 in the hybrid model that has a relatively fixed number of sessions likely arose due to the study capture period ending while some patients were early in their course of treatment. Technological barriers, the

most common reason for failure of virtual healthcare [10, 13, 14, 25], was less likely to be tolerated in the absence of an established therapeutic relationship, as indicated by our qualitative findings.

The different findings on patient satisfaction and adherence with virtual healthcare from prior meta-analyses [11, 24, 27, 28] and randomised trials [9, 12] compared to our study are likely due to the pragmatics of virtual healthcare delivery in real life, unlike in highly structured randomised trials. Previous research on virtually delivered interventions often provided patients with the equipment needed to complete the intervention remotely and successfully adhere to the intervention [24, 26]. As highlighted by our qualitative findings, this is something unlikely to happen in real life when healthcare is delivered completely virtually. The large number of people excluded from randomised trials (e.g. [9, 12]) and more pragmatic observational studies [10, 25] of virtually delivered rehabilitation, usually owing to lack of access to adequate technology [13, 14, 25], further reinforces our findings that virtual delivery of physiotherapy and exercise physiology services may suit some but not all patients. Indeed, evidence from discrete choice experiments designed to elicit consumer preference demonstrate lack of preference for fully virtual models [27]. Taken together, these results explain why a hybrid model combining some in-person consultations with some virtual consultations are associated with good outcomes, as seen in our previous work [13], and may be a better solution to increase access to care. Ultimately, offering patients choice in how they access care may improve patient outcomes and treatment adherence [29].

From a delivery perspective, we found that clinicians were overall confident in their communication and technical skills to successfully deliver virtual healthcare [30]. Similar to prior research, clinician confidence was higher among those who developed and used their skillset in delivering virtual healthcare on a regular and ongoing basis [30]. This suggests the need for clinicians who will provide care virtually to maintain a regular caseload of such patients, rather than providing virtual care on an ad hoc basis. Interestingly, we and others found that clinical competence in successfully delivering healthcare virtually required both professional expertise [30] and a willingness to be flexible in their approach to work [31].

This study was limited by a small sample size which limited our ability to directly compare in-person, hybrid and fully online service delivery models. As this study analysed real-world service provision, clinicians treated a wide variety of patients according to their clinical expertise. While the lack of strict patient inclusion/exclusion criteria or protocol-led interventions may limit inference about the suitability of a particular service delivery model according to specific interventions or patient populations, our findings reflect real-life challenges encountered by clinicians working in general clinics who treat a wide variety of patients. This is particularly true of rural and remote areas where there is difficulty accessing specialist care [7].

In conclusion, the results of this study, combined with prior findings by us [13] and others [11, 25], suggest that there may be a place for virtual healthcare to help fill access gaps in rural areas. However, care needs to be taken to identify patients who would benefit most from such models, as high uptake of virtual healthcare in rural areas is not always guaranteed [22]. Certain populations, eg, people with complex balance and/or cognitive impairment [11, 13] and people unfamiliar with technology [25, 32], appear not to benefit from virtually delivered exercise interventions. Poor internet and/or mobile phone connectivity in more remote areas may further impact uptake of virtual healthcare [13, 33]. While some populations, eg, musculoskeletal, post-operative and cardiac patients [23, 24, 28], are likely to adopt and benefit from virtual care, our results highlight the importance of establishing a therapeutic relationship and ensuring people have access to adequate resources, including access to and familiarity with technology as well as access to appropriate exercise equipment. Our findings are consistent with and reinforced in recently published guidelines for embedding virtual care into clinical practice

[34]. At the population level, a hybrid model incorporating limited in-person consultations with a number of virtual consultations [13, 27] appears a more viable option than a completely virtual model for addressing access issues in rural areas. More research is needed to learn the ideal dose of in-person and virtual consultations to maintain clinical engagement whilst being geographically and economically sustainable in rural settings.

## Supporting information

**S1 File. Supplementary material 1.** Patient interview and clinician focus group interview guides.
(DOCX)

**S2 File. Minimal data set.**
(XLSX)

## Acknowledgments

We would like to thank Taylah Duncan for extracting the de-identified data for analysis, and the staff and patients of Phyz X and Phyz X 2U who participated in the qualitative study.

## Author Contributions

**Conceptualization:** Serene S. Paul, Angela Hubbard, Justin Johnson, Sarah M. Dennis.

**Data curation:** Serene S. Paul, Angela Hubbard, Justin Johnson, Sarah M. Dennis.

**Formal analysis:** Serene S. Paul, Sarah M. Dennis.

**Funding acquisition:** Serene S. Paul.

**Investigation:** Serene S. Paul, Angela Hubbard, Justin Johnson.

**Methodology:** Sarah M. Dennis.

**Supervision:** Sarah M. Dennis.

**Writing – original draft:** Serene S. Paul.

**Writing – review & editing:** Angela Hubbard, Justin Johnson, Sarah M. Dennis.

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
