## [Decision Letter · Decision Letter 0]

21 Feb 2022

PONE-D-21-11529

Transition to a virtual model of physiotherapy and exercise physiology in response to COVID-19 for people in a rural Australia: is it a viable solution to increase access to allied health for rural populations?

PLOS ONE

Dear Dr. Dennis,

Thank you for submitting your manuscript to PLOS ONE. After careful consideration, we feel that it has merit but does not fully meet PLOS ONE’s publication criteria as it currently stands. Therefore, we invite you to submit a revised version of the manuscript that addresses the points raised during the review process.

The manuscript has been evaluated by two reviewers, and their comments are available below. The reviewers have raised a number of concerns that need attention. They request additional information on methodological aspects of the study, results and conclusions. 

Could you please revise the manuscript to carefully address the concerns raised?

We look forward to receiving your revised manuscript.

Kind regards,

Elisa Panada

Associate Editor

PLOS ONE

Journal Requirements:

2. Please provide additional details regarding participant consent. In the ethics statement in the Methods and online submission information, please ensure that you have specified how verbal consent was documented and witnessed.

4. Thank you for stating the following in the Acknowledgments Section of your manuscript: "This study was supported by a Charles Perkins Centre (University of Sydney) Early Mid-Career Researcher 2020 Seed Funding grant (SP). The funders had no role in study design, data collection and analysis, decision to publish, or preparation of the manuscript. We would also like to thank Taylah Duncan for extracting the de-identified data for analysis, and the staff and patients of Phyz X and Phyz X 2U who participated in the qualitative study.

Conflict of interest: Serene Paul and Sarah Dennis have declared that no competing interests exist. Phyz X and Phyz X 2U constitute the practice reported in this study. We have read the journal's policy and the authors of this manuscript have the following competing interests. Angela Hubbard is a co-owner of Phyz X 2U. Justin Johnson owns Phyz X and co-owns Phyz X 2U."

"This study was supported by a Charles Perkins Centre (University of Sydney) Early Mid-Career Researcher 2020 Seed Funding grant (SP). The funders had no role in study design, data collection and analysis, decision to publish, or preparation of the manuscript. "

5. Thank you for stating the following in the Competing Interests section: "Conflict of interest: Serene Paul and Sarah Dennis have declared that no competing interests exist. Phyz X and Phyz X 2U constitute the practice reported in this study. We have read the journal's policy and the authors of this manuscript have the following competing interests. Angela Hubbard is a co-owner of Phyz X 2U. Justin Johnson owns Phyz X and co-owns Phyz X 2U."

Reviewers' comments:

Reviewer's Responses to Questions

**Comments to the Author**

1. Is the manuscript technically sound, and do the data support the conclusions?

Reviewer #1: Partly

Reviewer #2: Partly

2. Has the statistical analysis been performed appropriately and rigorously? 

Reviewer #1: I Don't Know

Reviewer #2: Yes

3. Have the authors made all data underlying the findings in their manuscript fully available?

Reviewer #1: Yes

Reviewer #2: Yes

4. Is the manuscript presented in an intelligible fashion and written in standard English?

Reviewer #1: Yes

Reviewer #2: Yes

5. Review Comments to the Author

Reviewer #1: Thank you for allowing me to review this important and timely work. Below are suggested major revisions to increase the clarity and impact of this work.

ABSTRACT

1. In Line 5 can you clarify that this is a rural practice consisting of physiotherapy and exercise physiology providers? As worded, it appears there are only 2 providers at the practice.

2. Line 6, please state how was the virtual model of care delivered (e.g., phone, video, any integration with apps)?

3. Please clarify whether exercise physiologists are considered clinicians in Australia and adjust the wording throughout the manuscript accordingly.

4. Line 9, please state the kind of mixed-methods approach utilized. It appears to be convergent but will need clarity.

5. Line 9-13 is a lengthy sentence and a needs restructuring for clarity (e.g., as written the monthly surveys are extracted from the database, is that correct?)

6. Line 20-21, it may not be appropriate to directly compare satisfaction of in-person to virtual to patients who may not have experienced both. This comment extends throughout the manuscript. My thought is that this data is descriptive.

7. Line 25-26, the themes as outlined do not make sense as standalones without context (e.g., appropriateness of the program, things occurring during treatment, communication). Please add more clarity and description.

INTRODUCTION

8. Line 51, can you clarify which conditions? It would be helpful to know if this was a controlled factor in the historical controls.

9. Lines 64-65, please re-word as this study did not directly measure access and equity.

METHODS

10. Line 72, please state the kind of mixed methods approached utilized.

11. Lines 82-83 appear missed place in the first paragraph briefly outlining the design.

12. Please add a section outlining the program delivery (mode, platform, length of sessions, rough structure of sessions, technology support/infrastructure for troubleshooting)

13. Please add the length of the interviews/focus groups and conceptual model guiding the interview questions (or more detail on how you developed interview/focus questions and probes). It may be helpful to add the interview guide to the supplementary/appendix material.

14. Please clarify whether the two focus groups were separated by discipline of mixed.

15. Lines 129-136 would be more appropriate in the first paragraph of the methods section.

16. Line 135, from the description of the clinical survey it did not appear that any questions were directed at “readiness” but more confidence/self-efficacy. Please clarify.

17. Were demographic data and characteristics collected on the providers? May give context when triangulating quant and qual data.

18. Line 142-143, it may be appropriate to disclose in the results how many patient satisfaction scores were reported as “no change” in the final data.

19. Lines 143-145, per my comment previously, it may not be appropriate to directly compare satisfaction of in-person to virtual to patients who may not have experienced both.

20. Line 155 please include mention of the de-brief in the methods section and not analysis.

21. Please provide additional detail on whether a codebook was created, how many researchers participated, how many discussions to reach a consensus. Did you complete any form of member checking to ensure accuracy of the interpretations? It may be helpful to add the codebook to the supplementary/appendix material.

22. The data analysis section needs a section on the mixed methods analysis and how you triangulated the data.

RESULTS

23. Please restructure the results staring with a section on “Patient Characteristics” and then the Qualitative Themes with quantitative data integrated within (as per a mixed methods approach). This section requires extensive restructuring to display as a mixed methods study. As it stands, the qualitative and quantitative findings do not appear integrated.

24. Please provide context to what constitutes “inner regional” and whether this is operationalized as rural.

25. Line 248, the theme “appropriateness of the program” is very unclear in the context. Would adaptation of the program elements be a better (and clearer) theme?

26. In addition, the themes need more description. For example, “patient complexity” as a stand alone theme isn’t as clear as “patient complexity influences success of virtual care”

27. Line 274, please add whether all patients received the app and what it consists of in the methods.

DISCUSSION

28. Lines 393-394, access was not measured explicitly. Please revise.

29. Lines 428-433, long sentence that needs restructuring for clarity.

CONCLUSION

30. Lines 464, again careful wording and pronouncement that this study addressed access when access/equity were not measured (unclear whether qualitative interviews addressed these questions)

Reviewer #2: COVID-19 pandemic, pointed as the greatest sanitary challenge in the 21st century, has demanded adjustments in behaviors and practices in order to avoid the spread of the disease. Social distancing measures, which represent one of the main strategies to fight coronavirus, challenge daily lives of billions of people worldwide, and directly affect the organization of healthcare services.

Telehealth during the COVID-19 pandemic meets the indispensable measures, acknowledged all over the world, to curb the disease transmission, such as the use of masks, along with social distancing.

Telehealth and Telemedicine evolve as potentially capable tools to ensure health care in a scenario where avoiding people’s mobility is deemed necessary, without hindering service, treatment and monitoring of patients online, fundamentally, those neediest populations who live in remote areas.

In the first epidemiological weeks of the pandemic, little was known about the signs, symptoms and proper management of the SARS-CoV-2 virus. The frequent update on the circulating disease was fundamental for the individuals as a whole, more specifically, for the professionals of multidisciplinary healthcare teams.

Therefore, Telehealth has gained visibility as a feasible alternative for the traditional service, contributing directly to the improvement in the quality of the health care, as well as to the access to consultations and treatment in remote areas.

Telehealth has contributed to the reduction of professionals and patients’ exposure to the virus, once this method of consultations and treatment may be held at home. In times of pandemic, it is an excellent tool, which helps save lives by means of touting quality information, and using established, remotely referred protocols.

Telehealth is a type of health care, providing quick response to the crisis, with innumerable benefits to its users. It surely assures healthcare professionals to exercise their profession without exposing themselves to hazardous situations.

Thus, reviewing the article titled “Transition to a virtual model of physiotherapy and exercise physiology in response to COVID-19 for people in a rural Australia: is it a viable solution to increase access to allied health for rural populations?”, I state some considerations below.

The addressed theme is relevant for the scientific, technological and innovation development. The introduction is well underpinned by updated references. The objective is clear, well defined, and its methodology is according to the proposed study design. I request the application of the confidence interval (CI/95%), which not only informs the variability/dispersion of point estimates, but also the confidence intervals may express the statistical significance of comparative tests.

In the results (line 164), the authors stated that there were 90 patients interested in accessing remote healthcare services. From those patients, 56% were new referrals. In line 165, the authors mentioned that 67% of the patients agreed with the remote service, while 33% chose to wait for the traditional service. The paragraph is confusing. I request the authors to clarify the reality of the remote healthcare services.

The authors did not describe the justifications/reasons for the reduction of patients’ satisfaction with the virtual program (lines 170, 171). With the halt in the COVID-19 restrictions, the authors mentioned the increase in the traditional healthcare services, and a consequent fall in the remote services. Therefore, what was the percentage of patients quitting the virtual services, as well as the patients and healthcare professionals’ justification for that?

Regarding the patients, suffering from varied musculoskeletal and neurological disorders (line 201/Table 2), in what way were the virtual exercises prescribed? What was the control and understanding level of those patients, and the proposed exercises? What was the percentage of adherence to the treatment among those patients? How many absences should the patient have to be excluded from the program? How was this control carried out in order to conduct the remote program of exercises? Was a family member present during the exercise prescription and follow-up? How was the patients’ safety control held regarding balance and fall prevention? How was the control for the correct execution of the prescribed exercise? Did any patients quit? If so, what was the percentage? All those questions should be responded in the research results.

In relation to the equipment, did the patients know about the new technologies? If they didn’t, how was it solved? Regarding the connection to access the virtual service in remote, rural areas, how was this issue managed? In what ways were those issues managed? In what ways did they affect the remote program? (lines 248-253).

The manuscript aims to think over the adherence to virtual healthcare services of patients living in remote areas; to compare virtual and traditional healthcare services; to profile the patients who accessed virtual healthcare services; to verify the proportion of adherence to the virtual service by health professionals and patients, and, finally, to assess the availability and acceptance of the remote model.

The implementation of any new technological initiatives requires planning, resources and, obviously, funding. Challenges have been observed and reported by the authors of this article, who observed low patient adherence to this type of healthcare service. Some of those challenges, and the most important ones are the lack of knowledge of such technologies on the part of the population who lives in those remote areas, the access to the technology, the coverage of health insurance plans and government funding. Another important point is to establish a receptive relationship between therapist and patient.

Some issues were pointed out, showing the need to clarify the obtained result. The fact that calls attention to the established program is that patients, with more vulnerable health conditions (musculoskeletal and neurological disorders) were performing exercises in a remote way, without follow-up, at risk of performing them wrongly, prone to falls as well. It is necessary to demonstrate how the virtual Program of exercises works, the protocol(s) used, how the approach/referral/types of exercises were held in face of diverse etiologies.

Despite the interesting theme, I point out that the article requires several adjustments to be published.

The authors could compare 3 types of programs (traditional, hybrid and virtual), reporting the weaknesses and strengths of therapist and patient for each model, and describing the protocols used for each model.

6. PLOS authors have the option to publish the peer review history of their article (what does this mean?). If published, this will include your full peer review and any attached files.

Reviewer #1: No

Reviewer #2: No

---

## [Author Response · Author response to Decision Letter 0]

4 Apr 2022

Journal required revisions

Thank you. We have revised the manuscript to ensure it meets PLOS ONE’s style requirements and named all files correctly.

2. Please provide additional details regarding participant consent. In the ethics statement in the Methods and online submission information, please ensure that you have specified how verbal consent was documented and witnessed.

We have updated the ethics statement of the methods to state how consent was obtained (lines 99-102):

“The study received ethical approval from the University of Sydney Human Research Ethics Committee (reference: 2020/412). De-identified clinical data was obtained under a waiver of consent. Patients provided verbal informed consent to participate in the interviews, while the clinicians provided written informed consent to participate in the focus groups.”

We have ensured that information provided in the ‘Funding Information’ and ‘Financial Disclosure’ sections match. As there was no grant number for the award received, none is provided.

4. Thank you for stating the following in the Acknowledgments Section of your manuscript: "This study was supported by a Charles Perkins Centre (University of Sydney) Early Mid-Career Researcher 2020 Seed Funding grant (SP). The funders had no role in study design, data collection and analysis, decision to publish, or preparation of the manuscript. We would also like to thank Taylah Duncan for extracting the de-identified data for analysis, and the staff and patients of Phyz X and Phyz X 2U who participated in the qualitative study.

Conflict of interest: Serene Paul and Sarah Dennis have declared that no competing interests exist. Phyz X and Phyz X 2U constitute the practice reported in this study. We have read the journal's policy and the authors of this manuscript have the following competing interests. Angela Hubbard is a co-owner of Phyz X 2U. Justin Johnson owns Phyz X and co-owns Phyz X 2U."

We note that you have provided funding information that is not currently declared in your Funding Statement. However, funding information should not appear in the Acknowledgments section or other areas of your manuscript. We will only publish funding information present in the Funding Statement section of the online submission form. Please remove any funding-related text from the manuscript and let us know how you would like to update your Funding Statement. Currently, your Funding Statement reads as follows: 

"This study was supported by a Charles Perkins Centre (University of Sydney) Early Mid-Career Researcher 2020 Seed Funding grant (SP). The funders had no role in study design, data collection and analysis, decision to publish, or preparation of the manuscript."

We have updated the Funding Statement to the following and included it in the cover letter as requested: 

“This study was supported by a Charles Perkins Centre (University of Sydney) Early Mid-Career Researcher 2020 Seed Funding grant (SP). The funders had no role in study design, data collection and analysis, decision to publish, or preparation of the manuscript.”

We have removed all mention of funding and conflict of interest from the acknowledgements section. The acknowledgements section which now reads:

“We would like to thank Taylah Duncan for extracting the de-identified data for analysis, and the staff and patients of Phyz X and Phyz X 2U who participated in the qualitative study.”

5. Thank you for stating the following in the Competing Interests section: "Conflict of interest: Serene Paul and Sarah Dennis have declared that no competing interests exist. Phyz X and Phyz X 2U constitute the practice reported in this study. We have read the journal's policy and the authors of this manuscript have the following competing interests. Angela Hubbard is a co-owner of Phyz X 2U. Justin Johnson owns Phyz X and co-owns Phyz X 2U."

We have updated the Competing Interests statement in the cover letter as requested:

“Serene Paul and Sarah Dennis have declared that no competing interests exist. Phyz X and Phyz X 2U constitute the practice reported in this study. We have read the journal's policy and the authors of this manuscript have the following competing interests. Angela Hubbard is a co-owner of Phyz X 2U. Justin Johnson owns Phyz X and co-owns Phyz X 2U. This does not alter our adherence to PLOS ONE policies on sharing data and materials.”

We have provided the minimal dataset as Supporting Information (S2 file. Minimal data set).

Review Comments to the Author

Reviewer #1: Thank you for allowing me to review this important and timely work. Below are suggested major revisions to increase the clarity and impact of this work.

ABSTRACT

1. In Line 5 can you clarify that this is a rural practice consisting of physiotherapy and exercise physiology providers? As worded, it appears there are only 2 providers at the practice.

Thank you, we have reworded this sentence (lines 4-7) so it now reads:

“The COVID-19 pandemic provided an opportunity for a rural practice of physiotherapists and exercise physiologists to transition service delivery to a virtual model of care which utilised a combination of phone, video, an exercise app and/or paper handouts.”

2. Line 6, please state how was the virtual model of care delivered (e.g., phone, video, any integration with apps)?

We have added this detail as requested (lines 4-7):

“The COVID-19 pandemic provided an opportunity for a rural practice of physiotherapists and exercise physiologists to transition service delivery to a virtual model of care which utilised a combination of phone, video, an exercise app and/or paper handouts.”

3. Please clarify whether exercise physiologists are considered clinicians in Australia and adjust the wording throughout the manuscript accordingly.

Exercise physiologists are considered clinicians in Australia. We have modified the wording in the manuscript to reflect this (lines 4-7 as per our response to comments 1-2 and lines 127-128):

“At the start of the lockdown, clinicians (physiotherapists and exercise physiologists) completed a monthly survey to track their confidence in delivering virtual health services.”

4. Line 9, please state the kind of mixed-methods approach utilized. It appears to be convergent but will need clarity.

We have modified the abstract to state the mixed methods approached used in this study (line 9):

“A parallel convergent mixed methods study was conducted.”

5. Line 9-13 is a lengthy sentence and a needs restructuring for clarity (e.g., as written the monthly surveys are extracted from the database, is that correct?)

Thank you for the helpful suggestion. We have rewritten this section to improve clarity (lines 9-14):

“De-identified data from patients who were offered the virtual service between 15 March 2020 and 30 September 2020 were extracted from the database of the rural practice, as were data from patients attending the practice in-person during the same time in 2019 to serve as a historical comparison. De-identified data from a monthly survey tracking clinician experiences of delivering care virtually was also obtained from the practice.”

6. Line 20-21, it may not be appropriate to directly compare satisfaction of in-person to virtual to patients who may not have experienced both. This comment extends throughout the manuscript. My thought is that this data is descriptive.

While true that not all patients would have accessed and therefore experienced both in-person and virtual allied health services, we feel it is important to understand how satisfied patients are with the care they receive, as this indicates the likelihood they will continue with care. We used independent samples t-tests to make this comparison, which is stated in the methods (lines 15 and 165-167); we have further clarified this point in Table 2’s legend (line 228).

For the abstract, we have reworded this sentence (lines 21-23):

“Patients attended fewer sessional appointments virtually and a smaller proportion of patients reported high satisfaction with virtual care, compared to those who received in-person care the previous year (p<.05).” 

7. Line 25-26, the themes as outlined do not make sense as standalones without context (e.g., appropriateness of the program, things occurring during treatment, communication). Please add more clarity and description.

Thank you, we have revised this sentence (lines 25-28) to improve clarity:

“Five themes influencing the success of virtual allied health provision emerged from patient interviews and clinician focus groups: adaptation of program elements for virtual delivery, things occurring during virtual treatment, clinician flexibility, patient and communication.”

INTRODUCTION

8. Line 51, can you clarify which conditions? It would be helpful to know if this was a controlled factor in the historical controls.

In Australia, lack of reimbursement for virtually delivered allied health sessions was not based on patients’ conditions, but rather on the provider’s profession. Prior to the pandemic, psychologists were the only allied health providers who could be reimbursed for providing virtual care by the Australian government funded Medicare program; physiotherapy and exercise physiology were among the other allied health providers who were not funded to provide care virtually. We have modified this sentence (lines 47-50) to improve clarity:

“In Australia, a further barrier to virtual care was that most health insurance plans and the government funded Medicare system did not cover virtual delivery of physiotherapy or exercise physiology interventions, adding to the access issues for people in rural and remote areas.”

9. Lines 64-65, please re-word as this study did not directly measure access and equity.

We have reworded this sentence (lines 65-65) to read:

“The results will inform whether a fully virtual healthcare model may be a feasible solution to help address poor access to allied healthcare beyond the COVID-19 pandemic.”

METHODS

10. Line 72, please state the kind of mixed methods approached utilized.

We have revised the methods to state the mixed methods approach used in this study (lines 72-74):

“We conducted a parallel convergent mixed methods study (17) comprising an observational study with a historical comparison and qualitative interviews using a phenomenological approach with patients and clinicians.”

11. Lines 82-83 appear missed place in the first paragraph briefly outlining the design.

Thank you for identifying this inconsistency. We have moved this sentence to the fourth paragraph of the methods (lines 127-128) where we describe the monthly clinician survey.

12. Please add a section outlining the program delivery (mode, platform, length of sessions, rough structure of sessions, technology support/infrastructure for troubleshooting)

We have added these details to the methods as requested (lines 79-98):

“In-person services began with an initial consultation which comprised a thorough assessment followed by intervention as determined by the clinician, this could include provision of advice/education, exercise and/or manual therapy; subsequent follow-up consultations were booked as required. Follow-up consultations comprised a brief assessment followed by intervention as indicated. The patient was discharged from the service when the patient and treating clinician determined that no further follow-up was required (i.e., sessional service). The mobile outreach service had a similar in-person initial consultation as the in-person services, with the final evaluation 12 weeks later following the same format. During the intervening 10 weeks, intervention was in the form of weekly health coaching delivered virtually to update and progress exercise programs and provide health advice/education. All consultations were 30 minutes in duration except for the mobile outreach service’s initial consultation which was 60 minutes. During the partial lockdown period of the COVID-19 pandemic the practice introduced a fully virtual health model and ceased the mobile outreach service, though those who lived close to the two rural towns were able to access the traditional in-person service delivery model, as healthcare was exempt from COVID-19 lockdown restrictions in Australia. The fully virtual model followed the format of either the sessional in-person service or the structured 12-week mobile outreach service, depending on the patient’s preference. The platforms used depended on the patient’s preference and access, and varied from phone to freely available videoconferencing software, with or without use of the exercise app (Physitrack, London, UK). The practice did not have formal technology support, hence any troubleshooting was conducted by the clinician prior to or during the consultation.”

13. Please add the length of the interviews/focus groups and conceptual model guiding the interview questions (or more detail on how you developed interview/focus questions and probes). It may be helpful to add the interview guide to the supplementary/appendix material.

We have added these details to the methods as requested (lines 72-74, 136-137 and 146-147):

“We conducted a parallel convergent mixed methods study (17) comprising an observational study with a historical comparison and qualitative interviews using a phenomenological approach with patients and clinicians.”

“Semi-structured interviews with patients who had taken part in the virtual program were conducted between August 2020 and January 2021; each interview lasted approximately 45 mins.”

“Focus groups were conducted with physiotherapists and exercise physiologists who provided virtual care during the study period; each focus group lasted approximately 60 mins.”

We have added the interview guides as supplementary material (line 153):

“Supplementary material 1 contains the interview and focus group guides.”

14. Please clarify whether the two focus groups were separated by discipline of mixed.

The two groups were separated by discipline. This occurred coincidentally due to the availability of the clinicians, rather than by design. We have added this detail to the methods (lines 147-150):

“At the time of the study, there were seven clinicians working at the service who were invited to participate in two focus groups; by happenstance one group comprised physiotherapists and the other group the exercise physiologists.”

15. Lines 129-136 would be more appropriate in the first paragraph of the methods section.

We have moved the study outcomes to the start of the methods section. Due to the extra details about service provision requested in response to comment 12, we have split this paragraph into two and added the outcomes to the end of the second paragraph (lines 102-108).

16. Line 135, from the description of the clinical survey it did not appear that any questions were directed at “readiness” but more confidence/self-efficacy. Please clarify.

Thank you, we have reworded “clinician readiness” to “clinician confidence” throughout the manuscript. 

17. Were demographic data and characteristics collected on the providers? May give context when triangulating quant and qual data.

Demographic data were not collected on the providers.

18. Line 142-143, it may be appropriate to disclose in the results how many patient satisfaction scores were reported as “no change” in the final data.

We have added this detail to the footnote of Table 2 (line 246):

“hn=5 with satisfaction unchanged were due to missing data at follow-up.”

19. Lines 143-145, per my comment previously, it may not be appropriate to directly compare satisfaction of in-person to virtual to patients who may not have experienced both.

While true that not all patients would have accessed and therefore experienced both in-person and virtual allied health services, we feel it is important to understand how satisfied patients are with the care they receive, as this indicates the likelihood they will continue with care. We used independent samples t-tests to make this comparison (as stated in the methods, lines 165-167) and we also clarified this point in Table 2’s legend (line 228).

20. Line 155 please include mention of the de-brief in the methods section and not analysis.

We have moved the section referring to rigour of the qualitative component to the methods section (lines 153-158) as requested.

21. Please provide additional detail on whether a codebook was created, how many researchers participated, how many discussions to reach a consensus. Did you complete any form of member checking to ensure accuracy of the interpretations? It may be helpful to add the codebook to the supplementary/appendix material.

We have added details of the researchers who conducted the interviews/focus groups to the methods (lines 154-156):

“Credibility was assured by ensuring that the interviews were conducted by researchers experienced in qualitative research (SD and SP) and there were debriefs after each interview or focus group between these two researchers.”

Member checking was conducted with the clinicians only. We have added this detail to the methods (lines 151-152):

“Member checking with the clinician participants was conducted by sharing the transcripts with them and seeking feedback on accuracy.”

We coded the data inductively and did not develop a formal codebook, rather the codes and themes were organised in a coding tree. We do not feel that adding the coding tree to the supplementary material will provide any further useful information.

22. The data analysis section needs a section on the mixed methods analysis and how you triangulated the data.

Thank you, we have added these details to the data analysis section (lines 178-183):

“Finally, the results of the quantitative and qualitative data were merged to inform and enrich the findings. The qualitative results provided explanation for some of the quantitative findings and enabled the researchers to develop a more detailed understanding of the experiences of virtual care. Data from the patient and clinician interviews were triangulated. The research team included experienced quantitative and qualitative researchers (SD, SP) and clinicians with experience of delivering virtual and in-person services and this may have influenced the interpretation of the data.”

RESULTS

23. Please restructure the results staring with a section on “Patient Characteristics” and then the Qualitative Themes with quantitative data integrated within (as per a mixed methods approach). This section requires extensive restructuring to display as a mixed methods study. As it stands, the qualitative and quantitative findings do not appear integrated.

We have revised the results section and provided more links between the quantitative and qualitative data. The quantitative data is provided separately for patients and clinicians and the qualitative data has triangulated the data from patients and clinicians; this makes it difficult to integrate more than we already have.

24. Please provide context to what constitutes “inner regional” and whether this is operationalized as rural.

We have updated the methods (lines 122-124) and the footnote of Table 1 (lines 214-215) to provide more clarity on the Australian classification of rural areas:

“place of residence (classified as metropolitan, inner regional, outer regional, remote, very remote; the latter four are rural areas with progressively smaller populations and greater distances from major population centres; the latter four are rural areas with progressively smaller populations and greater distances from major population centres) (18, 19)”

25. Line 248, the theme “appropriateness of the program” is very unclear in the context. Would adaptation of the program elements be a better (and clearer) theme?

Thank you for this helpful suggestion, we have modified the name of this theme to ‘adaptation of program elements for virtual delivery’.

26. In addition, the themes need more description. For example, “patient complexity” as a stand alone theme isn’t as clear as “patient complexity influences success of virtual care”

We have modified the abstract (lines 25-26) and results section (lines 276-277) to clarify that these themes influence the success of virtual allied health consultations. 

27. Line 274, please add whether all patients received the app and what it consists of in the methods.

We have added this detail to the methods (lines 95-97). Please also see our response to comment 12.

DISCUSSION

28. Lines 393-394, access was not measured explicitly. Please revise.

Thank you, we have reworded this sentence summarising our results (lines 415-419):

“Although COVID-19 provided an opportunity to evaluate to what extent virtual therapy was utilised to access allied health care in rural and remote areas, we found that there was a much lower uptake of virtual healthcare in comparison to patients who chose to either continue with in-person care or wait until lockdown restrictions eased fully, as healthcare was considered exempt from COVID-19 restrictions in Australia.”

29. Lines 428-433, long sentence that needs restructuring for clarity.

Thank you, we have revised this section to improve clarity (lines 445-451):

“Our qualitative results highlighted the importance of establishing a therapeutic relationship for virtually delivered healthcare to succeed. This is an important and new finding, as prior studies demonstrating the benefits of virtually-delivered exercise programs assessed physical outcomes in-person at baseline prior to delivery of the virtual intervention. These prior studies thereby provided an opportunity to establish a therapeutic relationship (10, 11, 23, 28), unlike our study where both assessments and intervention were delivered virtually. This result also likely explains our finding of a substantial reduction in the number of sessional consultations delivered virtually;”

CONCLUSION

30. Lines 464, again careful wording and pronouncement that this study addressed access when access/equity were not measured (unclear whether qualitative interviews addressed these questions)

Thank you. We have rephrased this sentence to allude to the possibility that our findings could help inform the access issue since we did not directly measure this construct (lines 486-488):

“In conclusion, the results of this study, combined with prior findings by us (13) and others (11, 25), suggest that there may be a place for virtual healthcare to help fill access gaps in rural areas.”

Reviewer #2: COVID-19 pandemic, pointed as the greatest sanitary challenge in the 21st century, has demanded adjustments in behaviors and practices in order to avoid the spread of the disease. Social distancing measures, which represent one of the main strategies to fight coronavirus, challenge daily lives of billions of people worldwide, and directly affect the organization of healthcare services.

Telehealth during the COVID-19 pandemic meets the indispensable measures, acknowledged all over the world, to curb the disease transmission, such as the use of masks, along with social distancing.

Telehealth and Telemedicine evolve as potentially capable tools to ensure health care in a scenario where avoiding people’s mobility is deemed necessary, without hindering service, treatment and monitoring of patients online, fundamentally, those neediest populations who live in remote areas.

In the first epidemiological weeks of the pandemic, little was known about the signs, symptoms and proper management of the SARS-CoV-2 virus. The frequent update on the circulating disease was fundamental for the individuals as a whole, more specifically, for the professionals of multidisciplinary healthcare teams.

Therefore, Telehealth has gained visibility as a feasible alternative for the traditional service, contributing directly to the improvement in the quality of the health care, as well as to the access to consultations and treatment in remote areas.

Telehealth has contributed to the reduction of professionals and patients’ exposure to the virus, once this method of consultations and treatment may be held at home. In times of pandemic, it is an excellent tool, which helps save lives by means of touting quality information, and using established, remotely referred protocols.

Telehealth is a type of health care, providing quick response to the crisis, with innumerable benefits to its users. It surely assures healthcare professionals to exercise their profession without exposing themselves to hazardous situations.

Thus, reviewing the article titled “Transition to a virtual model of physiotherapy and exercise physiology in response to COVID-19 for people in a rural Australia: is it a viable solution to increase access to allied health for rural populations?”, I state some considerations below.

We agree with the points the reviewer raised in their summary above. The initial Covid-19 lockdown provided us an opportunity to conduct a natural experiment on uptake and satisfaction of allied health care in a real-world clinical setting in rural Australia, where this was not previously available due to policy and funding restrictions. Please note that this study was not a planned, prospective controlled study with intervention and control groups.

1. The addressed theme is relevant for the scientific, technological and innovation development. The introduction is well underpinned by updated references. The objective is clear, well defined, and its methodology is according to the proposed study design. I request the application of the confidence interval (CI/95%), which not only informs the variability/dispersion of point estimates, but also the confidence intervals may express the statistical significance of comparative tests.

Thank you for the positive review. As requested, we have included the mean difference and 95% confidence intervals of all comparisons in Tables 2 (as footnotes) and 3 (within the table).

2. In the results (line 164), the authors stated that there were 90 patients interested in accessing remote healthcare services. From those patients, 56% were new referrals. In line 165, the authors mentioned that 67% of the patients agreed with the remote service, while 33% chose to wait for the traditional service. The paragraph is confusing. I request the authors to clarify the reality of the remote healthcare services.

We have clarified in the results (lines 187-193) the number of people who expressed interest in accessing the virtual service, those who proceeded to actually access the virtual service, and what proportion of each were new referrals:

“Between 15 March 2020 and 30 September 2020, there were a total of 90 patients who expressed interest in accessing the virtual service, and of those 50 (56%) were new referrals. Of these 90 patients, 30 (33%) chose to wait for the resumption of in-person services and 60 (67%) took up the virtual service. The latter included 37 (62%) new referrals to the virtual service, while the remainder were already attending the standard service and chose to transition to the virtual service during the COVID-19 lockdown. These 60 patients attended a total of 242 consultations over the study period, a mean of 5.0 (SD 3.2) per patient.”

3. The authors did not describe the justifications/reasons for the reduction of patients’ satisfaction with the virtual program (lines 170, 171). With the halt in the COVID-19 restrictions, the authors mentioned the increase in the traditional healthcare services, and a consequent fall in the remote services. Therefore, what was the percentage of patients quitting the virtual services, as well as the patients and healthcare professionals’ justification for that?

The practice did not collect reasons behind patients’ satisfaction scores. However, we have revised the qualitative results section to better integrate relevant quantitative findings, including where participants in the qualitative interviews discussed reasons behind reduced satisfaction with the virtual program (eg, lines 283-285; please also see our response to Reviewer 1’s comment 22). 

We do not have data about how many patients quit the service, as some patients accessed the service frequently, while others may have had longer intervals between sessions. However, we have revised the methods to include detail of when patients ceased the virtual service (lines 83-84):

“The patient was discharged from the service when the patient and treating clinician determined that no further follow-up was required (i.e., sessional service).”

Please note that although virtual services declined since the lockdown ended, some patients living in more remote areas may have decided to continue with the virtual service if they felt this was a more convenient, or indeed the only accessible, option for them.

4. Regarding the patients, suffering from varied musculoskeletal and neurological disorders (line 201/Table 2), in what way were the virtual exercises prescribed? What was the control and understanding level of those patients, and the proposed exercises? What was the percentage of adherence to the treatment among those patients? How many absences should the patient have to be excluded from the program? How was this control carried out in order to conduct the remote program of exercises? Was a family member present during the exercise prescription and follow-up? How was the patients’ safety control held regarding balance and fall prevention? How was the control for the correct execution of the prescribed exercise? Did any patients quit? If so, what was the percentage? All those questions should be responded in the research results.

This was a natural experiment using routinely collected clinical data to explore uptake and attendance of virtual physiotherapy / EP program. There were no exclusions and our interest was to see to what extent patients and clinicians would engage with virtual care, the outcomes and experience. We have provided detail about the different models of care (in-person sessional, hybrid 12-week program, virtual sessional or 12-week programs) delivered by the practice in the methods (lines 75-98; please also see our response to Reviewer 1’s comment 12). As this study was a pragmatic evaluation of current service delivery models, all decisions about exercise prescription, follow-up and discharge from the service were at the discretion of the treating clinician/s based on their clinical expertise, though patients could choose to cease accessing the service. We have included this detail in the methods (lines 83-84):

“The patient was discharged from the service when the patient and treating clinician determined that no further follow-up was required (i.e., sessional service).”

Internet connectivity issues in rural and remote areas are an issue and patients used whatever they had access to, for some this was a landline telephone. Exercises prescribed in the virtual model were at the clinicians’ discretion, and were tailored to the individual patient’s level of understanding, home environment and available equipment. As described in the qualitative results (eg, lines 285-287), clinicians adapted their exercise prescription in the virtual service to ensure patient safety. 

Unfortunately, the practice did not collect data on patient adherence to their prescribed programs, so we are unable to report this outcome.

5. In relation to the equipment, did the patients know about the new technologies? If they didn’t, how was it solved? Regarding the connection to access the virtual service in remote, rural areas, how was this issue managed? In what ways were those issues managed? In what ways did they affect the remote program? (lines 248-253).

Clinicians and patients used the platform/s the patients had access to and were comfortable using. We have added this detail to the methods (lines 95-98; please also see our response to Review 1’s comment 12):

“The platforms used depended on the patient’s preference and access to technology, and varied from phone to freely available videoconferencing software, with or without use of an exercise app (Physitrack, London, UK). The practice did not have formal technology support, hence any troubleshooting was conducted by the clinician prior to or during the consultation.”

6. The manuscript aims to think over the adherence to virtual healthcare services of patients living in remote areas; to compare virtual and traditional healthcare services; to profile the patients who accessed virtual healthcare services; to verify the proportion of adherence to the virtual service by health professionals and patients, and, finally, to assess the availability and acceptance of the remote model.

The implementation of any new technological initiatives requires planning, resources and, obviously, funding. Challenges have been observed and reported by the authors of this article, who observed low patient adherence to this type of healthcare service. Some of those challenges, and the most important ones are the lack of knowledge of such technologies on the part of the population who lives in those remote areas, the access to the technology, the coverage of health insurance plans and government funding. Another important point is to establish a receptive relationship between therapist and patient.

Some issues were pointed out, showing the need to clarify the obtained result. The fact that calls attention to the established program is that patients, with more vulnerable health conditions (musculoskeletal and neurological disorders) were performing exercises in a remote way, without follow-up, at risk of performing them wrongly, prone to falls as well. It is necessary to demonstrate how the virtual Program of exercises works, the protocol(s) used, how the approach/referral/types of exercises were held in face of diverse etiologies.

Despite the interesting theme, I point out that the article requires several adjustments to be published.

This was not a prospectively controlled study, rather it was a natural experiment of real-world clinical service delivery at a time when the service delivery model had to be altered to accommodate the Covid-19 pandemic. There was no time to plan what the service might look like as the lockdown happened quickly. Physiotherapy and exercise physiology practices had difficult decisions to make to ensure that they could still generate an income and continue to provide clinical services for their patients. Further research does need to be undertaken that prospectively compares the different models of care for different conditions and the associated outcomes. The value of this study is that it provided data on the real-life acceptability to patients and clinicians of a virtual service.

We have included these limitations raised by the reviewer in the discussion (lines 478-485):

“This study was limited by a small sample size which limited our ability to directly compare in-person, hybrid and fully online service delivery models. As this study analysed real-world service provision, clinicians treated a wide variety of patients according to their clinical expertise. While the lack of strict patient inclusion/exclusion criteria or protocol-led interventions may limit inference about the suitability of a particular service delivery model according to specific interventions or patient populations, our findings reflect real-life challenges encountered by clinicians working in general clinics who treat a wide variety of patients. This is particularly true of rural and remote areas where there is difficulty accessing specialist care (7).”

7. The authors could compare 3 types of programs (traditional, hybrid and virtual), reporting the weaknesses and strengths of therapist and patient for each model, and describing the protocols used for each model.

Owing to the small numbers, we were not able to compare the 3 types of programs. As mentioned in response to comment 4, this study was a pragmatic evaluation of existing services. Hence the interventions delivered across all 3 types of programs were at the treating clinician’s expertise and discretion. In terms of patient access to each type of program, that likely resulted from a combination of their personal preferences and availability of services, eg, in some remote areas only the hybrid (mobile outreach service) program was available prior to 2020.

We have mentioned this in the limitations section of the discussion (lines 478-485):

“This study was limited by a small sample size which limited our ability to directly compare in-person, hybrid and fully online service delivery models. As this study analysed real-world service provision, clinicians treated a wide variety of patients according to their clinical expertise. While the lack of strict patient inclusion/exclusion criteria or protocol-led interventions may limit inference about the suitability of a particular service delivery model according to specific interventions or patient populations, our findings reflect real-life challenges encountered by clinicians working in general clinics who treat a wide variety of patients. This is particularly true of rural and remote areas where there is difficulty accessing specialist care (7).”

---

## [Decision Letter · Decision Letter 1]

16 Nov 2022

PONE-D-21-11529R1Transition to a virtual model of physiotherapy and exercise physiology in response to COVID-19 for people in a rural Australia: is it a viable solution to increase access to allied health for rural populations?PLOS ONE

Dear Dr. Dennis,

Thank you for submitting your manuscript to PLOS ONE. After careful consideration, we feel that it has merit but does not fully meet PLOS ONE’s publication criteria as it currently stands. Therefore, we invite you to submit a revised version of the manuscript that addresses the points raised during the review process.

We look forward to receiving your revised manuscript.

Kind regards,

Pracheth Raghuveer, MD, DNB

Academic Editor

PLOS ONE

Journal Requirements:

Reviewers' comments:

Reviewer's Responses to Questions

**Comments to the Author**

1. If the authors have adequately addressed your comments raised in a previous round of review and you feel that this manuscript is now acceptable for publication, you may indicate that here to bypass the “Comments to the Author” section, enter your conflict of interest statement in the “Confidential to Editor” section, and submit your "Accept" recommendation.

Reviewer #1: All comments have been addressed

Reviewer #3: (No Response)

2. Is the manuscript technically sound, and do the data support the conclusions?

Reviewer #1: Yes

Reviewer #3: Yes

3. Has the statistical analysis been performed appropriately and rigorously? 

Reviewer #1: Yes

Reviewer #3: I Don't Know

4. Have the authors made all data underlying the findings in their manuscript fully available?

Reviewer #1: Yes

Reviewer #3: Yes

5. Is the manuscript presented in an intelligible fashion and written in standard English?

Reviewer #1: Yes

Reviewer #3: Yes

6. Review Comments to the Author

Reviewer #1: Thank you for your thought-out responses to the comments raised. I appreciate your work in this area and the foundation from which future work can consider equitable access to virtual care in rural populations.

Reviewer #3: General comments:

Thank you for this interesting themed article. This concept is important to capture data and provide evidence to support continued use of telehealth/virtual care, particularly in the rural setting to increase access to services. I would like to encourage the authors to continue research in this field following on with planned studies targeting improving virtual care services. Meaningful results can be useful in lobbying government for continued funding for services delivered virtually.

It is understandable this study was unplanned and reactive to the unexpected scenario of the lockdown and does have some limitations in regards to small sample size etc but provides useful themes and a base for further research. It could also be noted that some of the initial patient/clinician concerns of moving to a virtual model of care may now be less of a worry today as society has become more accepting and experienced with this mode of communication since the lockdowns.

It would be useful to include a definition of virtual healthcare in your introduction to highlight it is healthcare of a real patient delivered virtually/via telehealth (not a study examining virtual patients ie. Simulated patients for clinical training)

When considering the feasibility of virtual care and the methods use include video and/or phone, this adds a level of complexity to analysis. Although easiest to provide care in person, care and exercise instruction via video has some challenges, but care via phone is very difficult as the element of instruction or demonstration is removed. The authors might like to add clarity around proportion of phone vs video consultations, and if phone was only used for interviewing not conducting an exercise session with the patient or if this influenced clinician confidence.

In your discussion, you might like to consider either via supplementary material or additional reference(s), how the concerns identified in the each of the themes could be addressed to improve the virtual experience for the patient/clinician. Eg theme: “adaptation of program element” .. have the governing bodies or clinic developed any guidelines for the specific safety considerations to be made when using virtual care? Are there any other guidelines or protocols that have since been developed that link with the themes or do your themes lead you to suggest guidelines that could/should be developed that would be of use to other clinicians?

Specific changes:

Line 57-65: Consider making minor rewording changes to this paragraph to clarify that this style of care was being used in rural/remote settings (and even city settings due to the lockdown) during the pandemic, not just the one clinic that was being studied.

Line 304: Theme titled “things occurring during treatment”, perhaps reconsider to avoid using “things” eg. Occurrences during virtual treatment

Thank you for considering the rural/remote population and investigating how to increase access to healthcare in both pandemic and non-pandemic settings.

7. PLOS authors have the option to publish the peer review history of their article (what does this mean?). If published, this will include your full peer review and any attached files.

Reviewer #1: No

Reviewer #3: No

---

## [Author Response · Author response to Decision Letter 1]

25 Nov 2022

Journal Required Revisions

We have reviewed our reference list and ensured that it is complete and correctly formatted. None of the references we have cited have been retracted. We have added three new references (references #29, #33 and #34) to the revised manuscript to address reviewer comments, as detailed below. 

Review Comments to the Author

Reviewer #1: Thank you for your thought-out responses to the comments raised. I appreciate your work in this area and the foundation from which future work can consider equitable access to virtual care in rural populations.

Thank you for the overall positive comment.

Reviewer #3: General comments:

Thank you for this interesting themed article. This concept is important to capture data and provide evidence to support continued use of telehealth/virtual care, particularly in the rural setting to increase access to services. I would like to encourage the authors to continue research in this field following on with planned studies targeting improving virtual care services. Meaningful results can be useful in lobbying government for continued funding for services delivered virtually.

Thank you for the overall positive comment.

It is understandable this study was unplanned and reactive to the unexpected scenario of the lockdown and does have some limitations in regards to small sample size etc but provides useful themes and a base for further research. It could also be noted that some of the initial patient/clinician concerns of moving to a virtual model of care may now be less of a worry today as society has become more accepting and experienced with this mode of communication since the lockdowns.

Thank you, this is true. However, while society has become more experienced with virtual communication, virtual provision of care is not always feasible, preferred or delivered well, as indicated by our results. Providing patients with the choice of in-person or virtual care is important, particularly for rural patients where access barriers are known issues. We have added this point to the Discussion (lines 491-492):

“Ultimately, offering patients choice in how they access care may improve patient outcomes and treatment adherence (29).”

It would be useful to include a definition of virtual healthcare in your introduction to highlight it is healthcare of a real patient delivered virtually/via telehealth (not a study examining virtual patients ie. Simulated patients for clinical training)

We have added a definition of virtual healthcare to the Introduction (lines 53-54):

“Virtual healthcare, ie, healthcare delivered via telephone or video, has the potential to improve access to allied health services for people living in rural areas (8).”

When considering the feasibility of virtual care and the methods use include video and/or phone, this adds a level of complexity to analysis. Although easiest to provide care in person, care and exercise instruction via video has some challenges, but care via phone is very difficult as the element of instruction or demonstration is removed. The authors might like to add clarity around proportion of phone vs video consultations, and if phone was only used for interviewing not conducting an exercise session with the patient or if this influenced clinician confidence.

Thank you for highlighting this issue. We have included in the Results the proportions of video and phone consultations over the study period (lines 275-276), both of which were highly utilised: 

 “Clinicians delivered virtual healthcare primarily through video and/or phone (93% and 85%, respectively, across the study period); the mode was determined by the technology available to the patient.”

From the dataset we were unable to ascertain which type of virtual consultation was used for assessment versus intervention, or whether the type of consultation influenced clinician confidence. From some of the raw qualitative data we ascertained that the type of virtual consultation provided was determined by patient preference, which were influenced by logistics (eg, lines 335-340), patient familiarity with technology and internet connectivity. We have addressed these points in the Conclusion (lines 512-517):

“However, care needs to be taken to identify patients who would benefit most from such models, as high uptake of virtual healthcare in rural areas is not always guaranteed (22). Certain populations, eg, people with complex balance and/or cognitive impairment (11, 13) and people unfamiliar with technology (25, 32), appear not to benefit from virtually delivered exercise interventions. Poor internet and/or mobile phone connectivity in more remote areas may further impact uptake of virtual healthcare (13, 33).”

In your discussion, you might like to consider either via supplementary material or additional reference(s), how the concerns identified in the each of the themes could be addressed to improve the virtual experience for the patient/clinician. Eg theme: “adaptation of program element”. Have the governing bodies or clinic developed any guidelines for the specific safety considerations to be made when using virtual care? Are there any other guidelines or protocols that have since been developed that link with the themes or do your themes lead you to suggest guidelines that could/should be developed that would be of use to other clinicians?

Thank you for the suggestion to improve the experience of virtually delivered healthcare for both patients and clinicians, we have added these clinical implications to the Discussion. Specifically, we have outlined how some of the concerns identified in the themes of patient complexity (lines 450-456) and clinician flexibility (lines 497-501) should be addressed. In the Conclusion (lines 521-523) we have also referenced a recent clinical practice guideline (reference #34) published to assist clinicians embed virtual care into routine clinical practice; this guideline reinforced some of our findings.

These are the corresponding changes to the manuscript, listed in the order noted above:

 “This was reflected in our qualitative findings where clinicians and patients alike felt that those with more complex conditions received more intensive therapy in-person. Clinically, it appears that triaging patients to offer virtual care to those with less complex needs and treating those with more complex needs in person may improve patient outcomes.”

“Similar to prior research, clinician confidence was higher among those who developed and used their skillset in delivering virtual healthcare on a regular and ongoing basis (30). This suggests the need for clinicians who will provide care virtually to maintain a regular caseload of such patients, rather than providing virtual care on an ad hoc basis. Interestingly, we and others found that clinical competence in successfully delivering healthcare virtually required both professional expertise (30) and a willingness to be flexible in their approach to work (31).” 

“our results highlight the importance of establishing a therapeutic relationship and ensuring people have access to adequate resources, including access to and familiarity with technology as well as access to appropriate exercise equipment. Our findings are consistent with and reinforced in recently published guidelines for embedding virtual care into clinical practice (34).”

Specific changes:

Line 57-65: Consider making minor rewording changes to this paragraph to clarify that this style of care was being used in rural/remote settings (and even city settings due to the lockdown) during the pandemic, not just the one clinic that was being studied.

We have reworded this paragraph to clarify that virtual delivery of healthcare was used during Covid lockdowns across rural and metropolitan settings (lines 74-79):

“The COVID-19 pandemic required providers in both rural and metropolitan settings to offer patients their physiotherapy and exercise physiology services through a new service delivery model using only virtual health. For some providers, including the rural allied health provider studied herein, patients who lived close to the clinic and preferred to use the traditional in-person service during the pandemic could opt to do so.”

Line 324: Theme titled “things occurring during treatment”, perhaps reconsider to avoid using “things” eg. Occurrences during virtual treatment

We have reworded this theme to “conduct of virtual treatment” throughout the manuscript (lines 39-41, 298 and 324).

Thank you for considering the rural/remote population and investigating how to increase access to healthcare in both pandemic and non-pandemic settings.

Thank you for the overall positive comment.

---

## [Decision Letter · Decision Letter 2]

11 Jan 2023

Transition to a virtual model of physiotherapy and exercise physiology in response to COVID-19 for people in a rural Australia: is it a viable solution to increase access to allied health for rural populations?

PONE-D-21-11529R2

Dear Dr. Sarah Dennis

We’re pleased to inform you that your manuscript has been judged scientifically suitable for publication and will be formally accepted for publication once it meets all outstanding technical requirements.

Kind regards,

Pracheth Raghuveer, MD, DNB

Academic Editor

PLOS ONE

Additional Editor Comments (optional):

Reviewers' comments:

Reviewer's Responses to Questions

**Comments to the Author**

1. If the authors have adequately addressed your comments raised in a previous round of review and you feel that this manuscript is now acceptable for publication, you may indicate that here to bypass the “Comments to the Author” section, enter your conflict of interest statement in the “Confidential to Editor” section, and submit your "Accept" recommendation.

Reviewer #1: All comments have been addressed

Reviewer #3: All comments have been addressed

2. Is the manuscript technically sound, and do the data support the conclusions?

Reviewer #1: Yes

Reviewer #3: Yes

3. Has the statistical analysis been performed appropriately and rigorously? 

Reviewer #1: Yes

Reviewer #3: Yes

4. Have the authors made all data underlying the findings in their manuscript fully available?

Reviewer #1: Yes

Reviewer #3: Yes

5. Is the manuscript presented in an intelligible fashion and written in standard English?

Reviewer #1: Yes

Reviewer #3: Yes

6. Review Comments to the Author

Reviewer #1: Thank you for the opportunity to review this manuscript and your continued efforts to understand and translate findings related to the delivery of physiotherapy via virtual modalities.

Reviewer #3: Thank you for addressing all my previous comments and your interest and contribution to the field to enhance patient care

7. PLOS authors have the option to publish the peer review history of their article (what does this mean?). If published, this will include your full peer review and any attached files.

Reviewer #1: No

Reviewer #3: No

---

## [Editor Report · Acceptance letter]

13 Jan 2023

PONE-D-21-11529R2 

Transition to a virtual model of physiotherapy and exercise physiology in response to COVID-19 for people in a rural Australia: is it a viable solution to increase access to allied health for rural populations? 

Dear Dr. Dennis:

I'm pleased to inform you that your manuscript has been deemed suitable for publication in PLOS ONE. Congratulations! Your manuscript is now with our production department. 

Kind regards, 

on behalf of

Dr. Pracheth Raghuveer 

Academic Editor

PLOS ONE